METHODS

# Estimating error rates for single molecule protein sequencing experiments

**Matthew Beauregard Smith**[1,2,3]*, **Kent VanderVelden**[3], **Thomas Blom**[3], **Heather D. Stout**[2,3], **James H. Mapes**[3], **Tucker M. Folsom**[3], **Christopher Martin**[3], **Angela M. Bardo**[2,3]*, **Edward M. Marcotte**[1,2]*

**1** Oden Institute, The University of Texas at Austin, Austin, Texas, United States of America, **2** Department of Molecular Biosciences, The University of Texas at Austin, Austin, Texas, United States of America, **3** Erisyon Inc., Austin Texas, United States of America

* mbsmith93@utexas.edu (MBS); angela@erisyon.com (AMB); marcotte@utexas.edu (EMM)

**Data Availability Statement:** All code and data are available: whatprot (Baum-Welch) is downloadable from https://github.com/marcottelab/whatprot; sigproc_v2 (image processing) and pfit_v1

## Abstract

The practical application of new single molecule protein sequencing (SMPS) technologies requires accurate estimates of their associated sequencing error rates. Here, we describe the development and application of two distinct parameter estimation methods for analyzing SMPS reads produced by fluorosequencing. A Hidden Markov Model (HMM) based approach, extends whatprot, where we previously used HMMs for SMPS peptide-read matching. This extension offers a principled approach for estimating key parameters for fluorosequencing experiments, including missed amino acid cleavages, dye loss, and peptide detachment. Specifically, we adapted the Baum-Welch algorithm, a standard technique to estimate transition probabilities for an HMM using expectation maximization, but modified here to estimate a small number of parameter values directly rather than estimating every transition probability independently. We demonstrate a high degree of accuracy on simulated data, but on experimental datasets, we observed that the model needed to be augmented with an additional error type, N-terminal blocking. This, in combination with data pre-processing, results in reasonable parameterizations of experimental datasets that agree with controlled experimental perturbations. A second independent implementation using a hybrid of DIRECT and Powell's method to reduce the root mean squared error (RMSE) between simulations and the real dataset was also developed. We compare these methods on both simulated and real data, finding that our Baum-Welch based approach outperforms DIRECT and Powell's method by most, but not all, criteria. Although some discrepancies between the results exist, we also find that both approaches provide similar error rate estimates from experimental single molecule fluorosequencing datasets.

## Author summary

Diverse new technologies are being developed for single-molecule protein sequencing, capable of identifying and quantifying mixtures of proteins at the level of individual molecules. There are many biochemical challenges intrinsic to high-throughput studies of proteins at such high sensitivity arising from their heterogeneous chemistries, sizes, and

(DIRECT + Powell's) from https://github.com/marcottelab/robust-fluorosequencing-plaster; and fluorosequencing datasets are deposited on Zenodo at https://zenodo.org/record/8137155.

**Funding:** M.B.S. acknowledges support from a Computational Sciences, Engineering, and Mathematics graduate program fellowship. E.M.M. acknowledges support from Erisyon, Inc., the National Institute of General Medical Sciences (R35GM122480), the National Institute of Child Health and Human Development (HD085901), and the Welch Foundation (F-1515). The funders had no role in study design, data collection and analysis, decision to publish, or preparation of the manuscript.

**Competing interests:** I have read the journal's policy and the authors of this manuscript have the following competing interests: A.M.B. and E.M.M. are co-founders and shareholders of Erisyon, Inc., and are co-inventors on granted patents or pending patent applications related to single-molecule protein sequencing. A.M.B. serves on the board of directors and E.M.M. serves on the scientific advisory board. M.B.S., K.V., T.B., H.D.S., J.H.M., T.M.F., C.M., and A.M.B. are affiliated with Erisyon, Inc., as employees or shareholders. H.D.S. is currently employed by UT Austin with funding from a Sponsored Research Agreement from Erisyon, Inc.

abundances. Beyond these challenges, the technologies themselves involve complex multi-step analytical processes. Thus, in developing and optimizing these technologies, it is important to consider the accuracy of each step and to have reliable approaches for estimating these accuracies. We focus on one particular single-molecule sequencing technology known as flourosequencing. We report and validate two methods for simultaneously determining the error-rates of each of the various steps of the fluorosequencing process. These new error estimation techniques will help researchers to better interpret the effects of changes to the chemistry and sample preparation used in fluorosequencing so that these steps can be improved. Further, more accurate determination of error rates will aid in the creation of better tools for the interpretation of this data.

## Introduction

Proteins are key components of all living organisms, but their roles and functions, and in particular their quantities, processing, and modifications, cannot be fully determined from DNA and RNA sequencing alone. Advancements in protein identification and quantification are much sought after by researchers in the field of single molecule protein sequencing. SMPS technologies apply concepts from DNA and RNA sequencing to protein analysis—including imaging or nanopore-based assays, high parallelism, and single molecule detection—with the hope of achieving the high throughput and sensitivity offered by such approaches [1–5].

Notably, early generations of single molecule DNA sequencing technologies were marked by high error rates, but subsequent optimization allowed these error rates to be driven down. For example, the first commercially-released MinION R7 nanopore sequencers in 2014 from Oxford Nanopore Technologies generally showed error rates at or above 30% error (per nucleotide, per molecule) [6–9], but improvements in chemistry and software rapidly brought error rates down to approximately 15% within just a year or two [10], and continued optimization has reduced the error rates to their current levels of < 5% (e.g. [11,12]). One might reasonably expect a similar trajectory for SMPS technologies, highlighting the importance of accurately estimating the sequencing parameters and error rates to facilitate practical applications and ongoing optimization of the methods.

In the SMPS technique known as fluorosequencing (Fig 1) [13,14], a biological sample of peptides is acquired, such as from the proteolytic digestion of proteins. The peptides are then covalently labeled with fluorescent dyes on specific types (or groups) of amino acids. The peptides are then immobilized by their carboxy termini in a microscope flow cell and imaged by total internal reflection fluorescence (TIRF) microscopy. By alternating microscopy with performing Edman degradation chemistry [15], which removes a single amino-terminal (N-terminal) amino acid from each peptide on each sequencing cycle, a time series of images is obtained capturing sequence information for many peptides in a highly parallel and scalable fashion. Signal processing of the resulting images is performed to identify peaks, corresponding to peptides, and quantify their fluorescence across the time series, thus providing the primary raw sequencing reads, each comprising a time series of fluorescence intensities for a single peptide molecule across the course of the sequencing experiment. As these reads capture the sequence positions of the fluorescently labeled amino acids within each peptide, they can then be matched to a reference database to identify the most likely peptides, and thus proteins, in the sample. We previously described the algorithm whatprot to perform this peptide-read matching process to classify the reads, assuming the set of fluorosequencing parameters for the experiment [16].

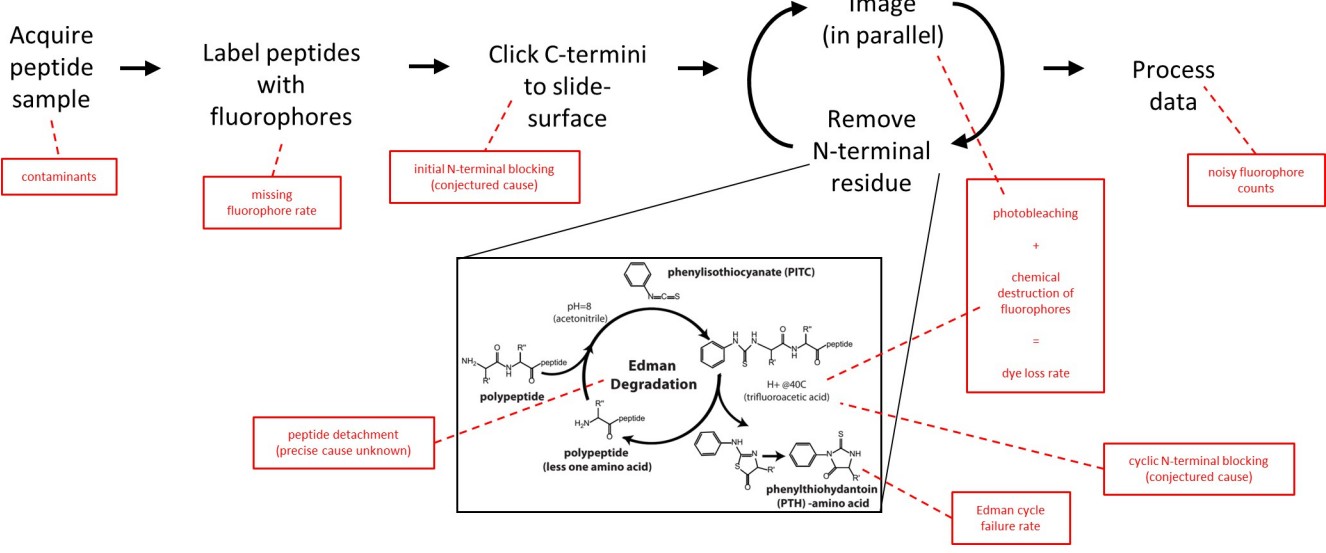

**Fig 1. Overview of single molecule protein fluorosequencing with various potential sources of errors highlighted in red.**

A sample of peptides is acquired, which due to biological factors or collection practices may contain contaminants. These peptides are then labeled, but this process is not expected to be 100% efficient, and a missing fluorophore rate must therefore be considered. Labeled peptides are clicked to the slide surface prior to sequencing; N-terminal modifications and non-specific attachment might potentially occur during this process. The peptides are then imaged and a single amino acid removed from each of their N-termini by Edman degradation, and these two processes iterated to fully sequence the peptides. Imaging photobleaches dyes, while incubation in trifluoroacetic acid (TFA) and phenyl isothiocyanate (PITC)/pyridine can result in chemical destruction of fluorophores. As photobleaching happens at a negligible rate in the imaging conditions used [13] and the dye loss rate is dominated by the contribution of chemical destruction, we combine these effects into a single dye loss rate. Edman degradation is not 100% efficient, and we model failures with the Edman cycle failure rate. We model the potential for blocking peptide N-termini during sequencing, which we call cyclic N-terminal blocking, and also model the detachment of intact peptides (e.g. by non-specific cleavage or washing off of non-specifically attached peptides). Finally, it can be difficult to entirely denoise the precise fluorophore counts due to overlapping intensity distributions, so we additionally recognize an error contribution from mis-assigned fluorophore counts.

The model analyzed here likely captures the most significant sequencing parameters, but may still omit elements. It has been developed over several years through a combination of intuition about the chemistry in use during sequencing and analysis of data from experimental runs, especially when there are large discrepancies between the real data and simulations based on fit parameters. Nonetheless, it seems inevitable that some forms of error may have been omitted and will need to be added into the models, which can be expanded as needed in the future.

Accurate estimation of the parameters of the model is helpful in two respects. Firstly, with better understanding of the underlying parameters, scientists working on improving the fluorosequencing technology can more easily draw intuitive connections between a change to the fluorosequencing process and the quality of the results during sequencing. Secondly, better estimation of parameters should improve the quality of classification of peptides and proteins,

improving the quality of sequencing analyses. Also, much as happens in other classification settings, we can anticipate that algorithms for interpreting single molecule protein sequencing data would benefit from being trained on larger datasets, such as those from simulations, provided they accurately reflect the sequencing process, and high quality error estimation will be critical for refining computational models of the sequencing process.

Whatprot, on which one of our parameter estimation methods is based, models the sequencing of each peptide by a distinct HMM, an example of which is illustrated in Fig 2. In general, HMMs consist of a set of states and a set of probabilistic transitions between those states.

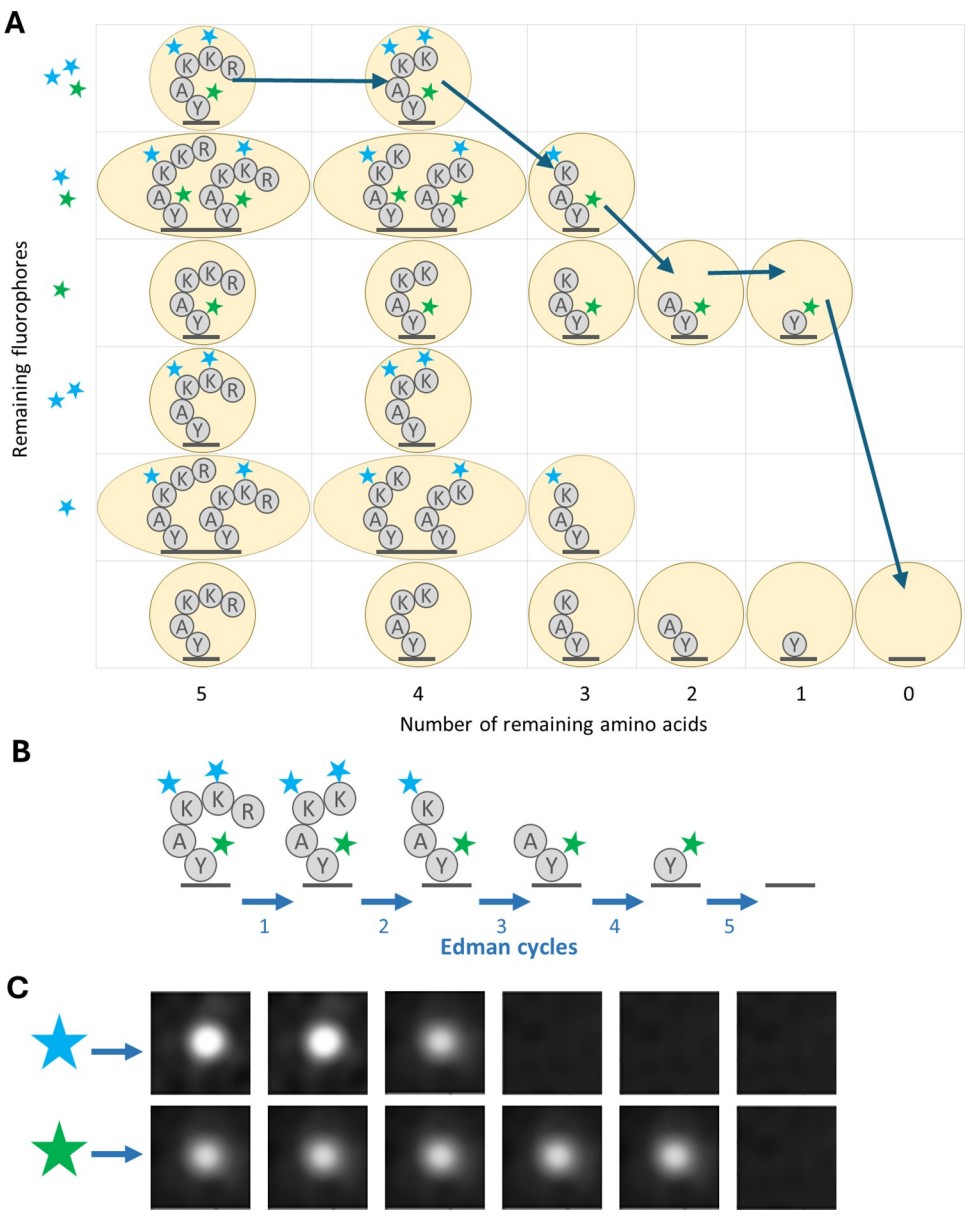

**Fig 2. Demonstration of the relationship between fluorosequencing data and the theoretical model.** (**A**) Illustration of a potential path through the states in the model, in this case the path represents perfect sequencing with no errors. (**B**) Illustration of the same peptide being sequenced with successive Edman cycles, aligned with the images in (**C**) the raw data we wish to analyze, i.e. the light emitted from a single molecule measured across two fluorescent channels (rows) for 5 Edman cycles (columns).

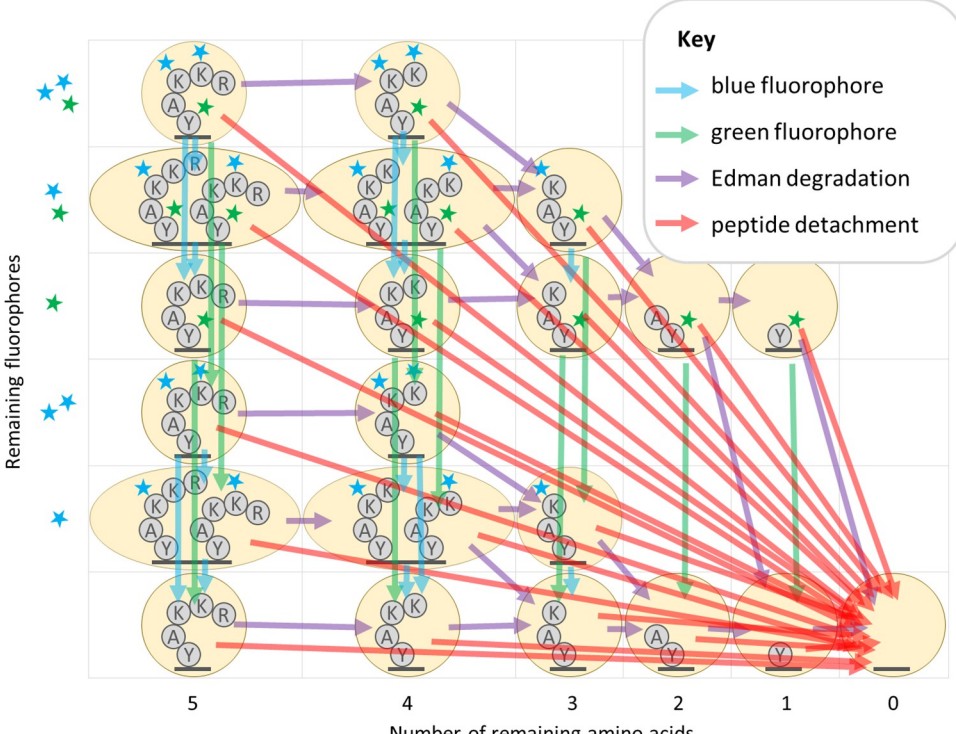

**Fig 3. Diagram of factored hidden Markov model for fluorosequencing of an example peptide with three labeled amino acids.**

The HMMs based on our model have a state for every possible condition of the peptide (Fig 2A), including the number of remaining amino acids (amino acids not yet removed by Edman degradation, as diagrammed in Fig 2B), and the combination of fluorophores still present on the peptide. Because each fluorophore can either be present and functioning or not, the number of states in the model grows exponentially with respect to the maximum possible number of fluorophores for the peptide given the experimental conditions. Transitions in our HMM (represented by arrows in our figures) can then represent the probability of moving from one state to another, based on the rates of error. Emissions then refer to the expected distribution of intensity as measured from the raw image data (Fig 2C).

This figure is adapted from Fig 7 in [16]. Arrows of a particular color indicate the non-zero entries in a factor of the transition matrix for a particular form of error (see key).

In [16] we demonstrated that significant optimizations of the HMM made the forward algorithm tractable to run even for large peptides with many fluorophores. In particular, we reduced the state space of the model by merging all states with the same number of successful Edman cycles and fluorophore counts of each color, and we also introduced a factorization of the transition matrix which, as a side effect, isolated the effects of each form of error from the others (Fig 3).

Separating forms of error proved fortuitous for parameter fitting, as it allows isolation of each error type. Further, we found that we could perform weighted maximum likelihood estimation, in a modification of the classical Baum-Welch algorithm [17], to estimate sequencing parameters directly. This approach offers some advantages over the classical Baum-Welch algorithm, including greater interpretability and direct ties to the underlying model of the

fluorosequencing process, and seems likely to be less prone to overfit the data since significantly less parameters are available to be fit.

We implemented these new capabilities in parameter estimation as an extension to *whatprot*. We demonstrate that the approach provides accurate estimates of the parameters used for simulated data, correctly identifies experimentally manipulated parameters in actual sequencing experiments, and gives similar parameter estimates to simpler, more general purpose, parameter estimation techniques that are by their nature less precise.

We also discuss a second independent implementation, using DIRECT [18,19] and Powell's method [20], reducing the root mean squared error (RMSE) between simulations and the real dataset, to provide an independent source of confirmation of results for real data.

Both methods provide a high degree of accuracy on simulated data, but on real datasets, we observed that the model needed to be augmented with an additional error type, N-terminal blocking. This results in reasonable parameterizations of experimental datasets that agree with controlled experimental perturbations.

A comparison of these methods on both simulated and real data demonstrates that our Baum-Welch based approach outperforms DIRECT and Powell's method by most, but not all, criteria. Although some discrepancies between the results exist, we also find that both approaches provide similar error rate estimates from experimental single molecule fluorosequencing datasets.

## Methods

### N-terminal blocking

In exploratory work for this chapter, we found that parameter estimation results on real datasets required that we consider an additional form of error that was not accounted for in our prior models of fluorosequencing: the N-terminus could have a chemical modification that blocks sequencing, which is either acquired before sequencing starts (initial N-terminal blocking) or while sequencing takes place (cyclic N-terminal blocking). To account for initial and cyclic N-terminal blocking, we adjusted our model, for purposes of Monte Carlo simulation, classification (as in [16]), and for parameter fitting.

We first doubled the number of states, in order that each of the former states now maps to two states, one with and one without N-terminal blocking. Edman degradation sub-transitions only apply to the states without N-terminal blocking and have no effect on the states with blocked N-termini. Other sub-transitions and the emission calculations are applied equivalently but separately to both sets of states as before. However, we require a new sub-transition factor of our transition computations to account for N-terminal blocking: with a parameterized probability, a state transition is made from a state to its N-terminally blocked duplicate. This new transition has two variants, one to account for initial N-terminal blocking, and another to account for per-cycle N-terminal blocking (Fig 4).

In addition to the unblocked states (top-left), we found we must also consider blocked states (bottom-right) in which Edman degradation is not possible. An additional transition matrix factor representing N-terminal blocking is needed to describe this behavior (yellow arrows).

### Ordinary Baum-Welch

We estimated parameters for fluorosequencing experiments using a modification to the classical Baum-Welch algorithm. First, we briefly review Baum-Welch [17], a standard technique to determine transition probabilities in an HMM. Dynamic programming is used to determine the probability, given a sequence of observed data, of being in each state at each time step. The probability of each particular transition occurring between each neighboring pair of time steps

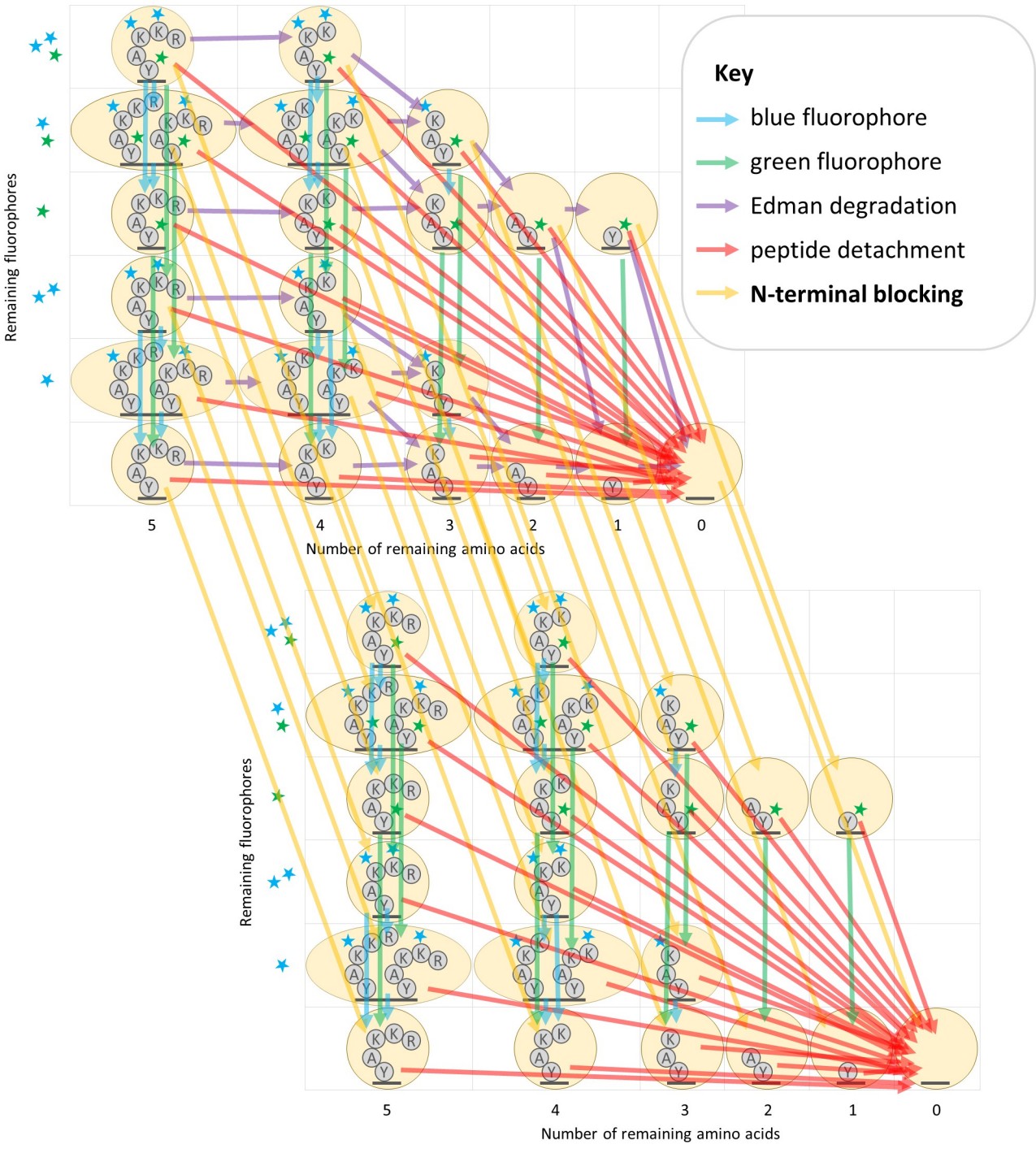

**Fig 4. Factored transition matrix diagram including N-terminal blocking.**

is also determined. Together, these probabilities allow a weighted maximum likelihood estimate (MLE) to be computed for every transition probability in the model.

Similarly, an estimate for the emission distribution can also be determined. This is done by simply computing a weighted MLE of the hypothetical distribution—in this case a normal distribution—in order to parameterize it, independently for each possible state.

We note that weighting this estimate is necessary to properly account for the fact that a state at one time step may be more likely, and therefore a better source of data, than that same step at a different time step; the more likely time step should have a greater impact on the final result. Additionally, in our model of a peptide undergoing fluorosequencing, many transitions are impossible; these transitions are omitted from the Baum-Welch procedure to improve runtime, as we know their probability to be zero.

Baum-Welch is a form of Expectation Maximization (EM) algorithm [21], and uses an existing estimate for modeling parameters to compute a better estimate of the modeling parameters. We therefore iterate this process until convergence is achieved. Additionally, Baum-Welch can be easily generalized to multiple independent and identically distributed sequences of output data, a simple but necessary generalization which we use in our work. This is discussed in more detail in S1 Appendix.

## Baum-Welch with isolation of error types

In [16] we improved the algorithmic complexity of the forward algorithm by factoring the transition matrix into a product of sparse matrices (Figs 3 and 5). We also showed that this factorization involved no loss in the accuracy of the model. For parameter estimation, only one peptide needs to be considered, so runtime considerations are of less significance. However, we find that this factorization serves a second purpose: isolation of our forms of error.

The Baum-Welch algorithm must be modified to take advantage of the factored transition matrix. In particular, we consider every step as an indexed series of sub-steps, each pertaining to a particular form of error; further, we can consider each transition as a series of sub-transitions, each pertaining to an application of one of our factors of the original transition matrix. We can then determine the probability of being in each of the states of the model at every sub-step of every time step, as well as the probability of any sub-transition between adjacent sub-steps. We then proceed as in the classical Baum-Welch with these probabilities to estimate the sub-transition probabilities using weighted MLEs (S2 Appendix).

This algorithm was never implemented, and is instead considered as a theoretical stepping stone to direct parameter estimation, described later in this document.

## Weighted maximum likelihood estimate for parameter estimation

While isolation of error types through factorization of the transition matrix should lead to a better model of fluorosequencing, it falls short of direct estimation of the model parameters. For this reason, we made a further modification to Baum-Welch. Instead of estimating each transition probability independently through weighted maximum likelihood estimation (MLE) in every iteration of Expectation Maximization, we use weighted MLE to directly estimate the parameters.

For example, consider the sub-transitions of dye loss for a particular color of fluorophore. If these sub-transitions are considered independently of each other, then their new probability for the next iteration of Expectation Maximization is given by the probability of the sub-transition, which is then normalized by dividing by the probability of being in the starting state at the sub-time step for the sub-transition. We instead consider each probabilistic transition as evidence for the value of an underlying parameter, in this case the per-cycle dye loss rate of the fluorophore.

Weighting of the MLE is necessary because $n$, the number of trials, and $x$, the number of successes (which for this application is counter-intuitively the probability of fluorophore loss, which is in a way a sort of failure), can only be determined probabilistically. However, Baum-Welch provides the weights for these probabilities. We can take $n$ to be a sum of the number of

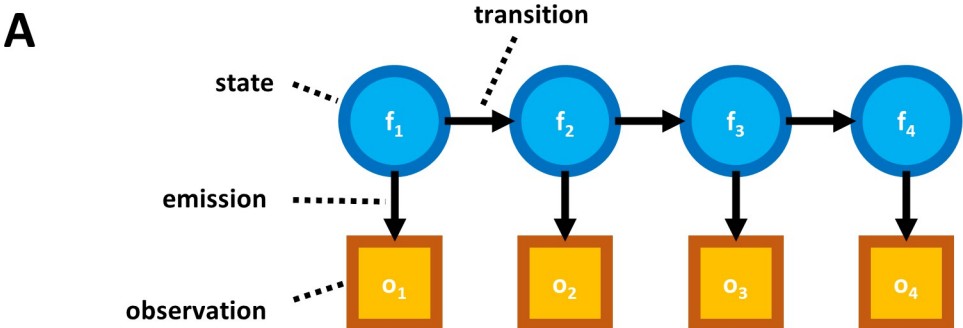

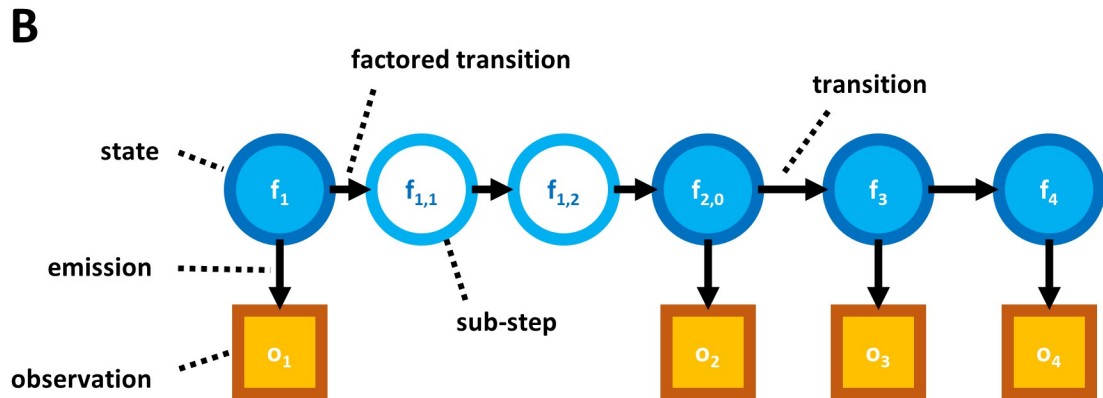

**Fig 5. Illustration of HMM factorization.** (**A**) Diagram of a non-factored HMM model. Arrows represent a conditional probability relationship. The transitions between states determine how a state at one time step is probabilistically related to the state at the preceding time step. Emissions represent how the observable data is probabilistically determined by the associated state. (**B**) Diagram including a factored transition. Breaking a transition into a factored product of sub-transitions introduces "sub-steps"; though not accurate models of any physical states of an actual peptide, these sub-steps prove useful for algorithmic purposes.

fluorophores for every state at the sub-time step before the sub-transition, weighted by the probabilities of being in each of those states. Similarly, we take $x$ to be a sum of the number of fluorophores of each destination state of a sub-transition, weighted by the probabilities of those sub-transitions.

This approach can be generalized to other parameters under consideration. This is done by mapping each starting state and sub-transition to a value which is then fed into a weighted estimate. In S3 Appendix, we provide a generalized formula for the MLE step of the Baum-Welch algorithm, along with descriptions of the value-mapping procedures used for each type of sub-transition.

## Bias correction for the missing fluorophore rate

The missing fluorophore rates (aka "dud rates", for brevity), which are the rates of non-fluorescing or absent fluorophores of each color before sequencing starts, have additional challenges. If all fluorophores on a peptide are missing, the peptide will not show up as a read in the sequencing results. This introduces a problematic bias, and if not accounted for in an MLE, the missing fluorophore rates may be estimated significantly lower than they should be (Fig 6). We derived an equation for the MLE of a binomial distribution with all instances of all fluorophores missing. Our MLE equation allows no closed-form solution that we know of; however, it does suggest an iterative method.

At least one functioning fluorophore must be present when sequencing starts to observe a peptide. This makes naive calculations of the missing fluorophore rate biased, and a correction is needed.

We interleave this iterative bias-correcting process with our implementation of the Baum-Welch algorithm. In particular, the missing fluorophore rates of the previous iteration determine the probability, $x$, that the peptide is missing from the dataset. We then compute $N(x/(1-x))$ where $N$ is the number of reads. This serves as an estimate of the number of missing reads, which we use for each missing fluorophore rate estimator as an instance of a peptide for which all of its dyes are missing.

The mathematical convergence properties are discussed with proofs in S4 Appendix.

## A straightforward alternative: DIRECT then Powell's method

This publication contains the first methods capable of estimating parameters for protein fluorosequencing that can properly account for interactions between different forms of error. Aside from synthetically generated datasets and experiments controlling specific factors, there are no baseline true values on real data for us to compare our results to. Naturally this makes it difficult to know whether the results can be trusted, so we felt it imperative to have a second method very different from the first. Agreement between the two methods then helps to improve confidence in the accuracy of both methods.

As a second method of parameter estimation and an independent check on the HMM-based approach, we considered a relatively simple alternative, in which we minimize an objective function: every read in the data is first reduced to a dye track, which is an integer approximation of the original read, where the integers represent the best estimates of the numbers of fluorophores of each color at each time step. Given a parameter estimation, we can then simulate dye tracks, and compute the root-mean square error (RMSE) between the counts of each dye track in the original data and the simulated result.

To minimize the RMSE, we used Powell's method [20], which allows us to optimize a function for which the derivatives cannot be computed. It proceeds by, along one dimension (parameter) at a time, finding the minimum, and repeating this for each dimension (parameter) until desired convergence criteria are achieved.

Empirically, we found that Powell's method often halted in local minima. Thus, we first used the DIRECT global optimization method (Fig 7A) [18,19] to identify suitable initial conditions for Powell's method (Fig 7B). DIRECT proceeds by repeatedly trying two test points above and below a given starting point along a particular dimension, and choosing the best among those and the original point. It then chooses another dimension and repeats until desired specificity of the result is achieved.

As an alternative to Baum-Welch, we also explored a more general purpose approach. (**A**) We first apply DIRECT to rapidly identify a region of the parameter space that is likely to contain the global optimum (red dots). DIRECT proceeds by iteratively comparing three points and using these results to further subdivide the search space, as shown. (**B**) We then apply Powell's method, which iteratively minimizes the objective function by changing one variable at a time.

We found this combination of DIRECT and Powell's method to be generally effective as an alternative to Baum-Welch, with the caveat of first having to independently estimate dye tracks, which may introduce changes relative to those considered during Baum-Welch.

## Contaminants and abnormal intensity distributions

While both approaches work well on simulated data, we observed two additional factors affecting actual experimental data: (1) contaminants in the sample that behave like fluorophores,

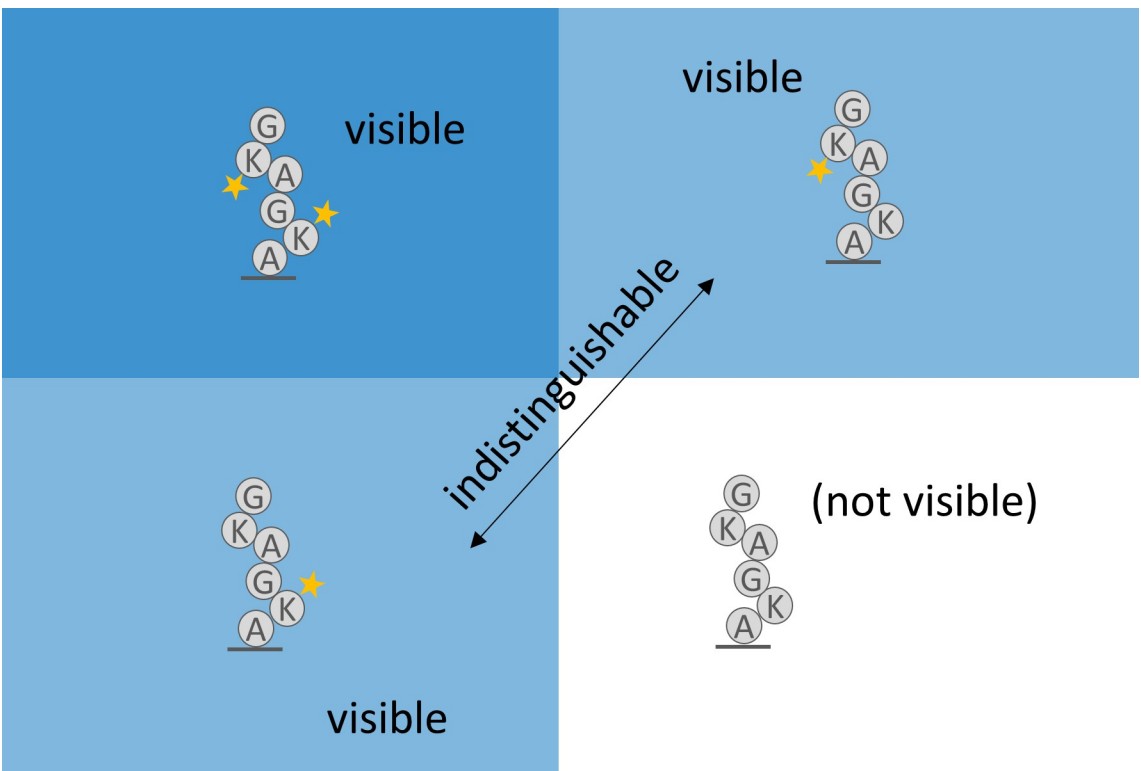

**Fig 6. Data is lost due to missing fluorophores.**

and (2) fluorophore intensity distributions not precisely fitting a normal distribution (see, for example, the histograms in Fig 8A).

Our first approach, applicable only to our Baum-Welch based method, was to truncate the normal distribution, as we did in [16] as a pruning technique. There, we employed this as a pruning operation to speed up the runtime of the forward algorithm, but reusing this technique to ignore highly unlikely data was ineffective in this context. Instead, we considered two additional approaches, which we use together for greater data integrity. These have the added benefit of compatibility with both of our methods.

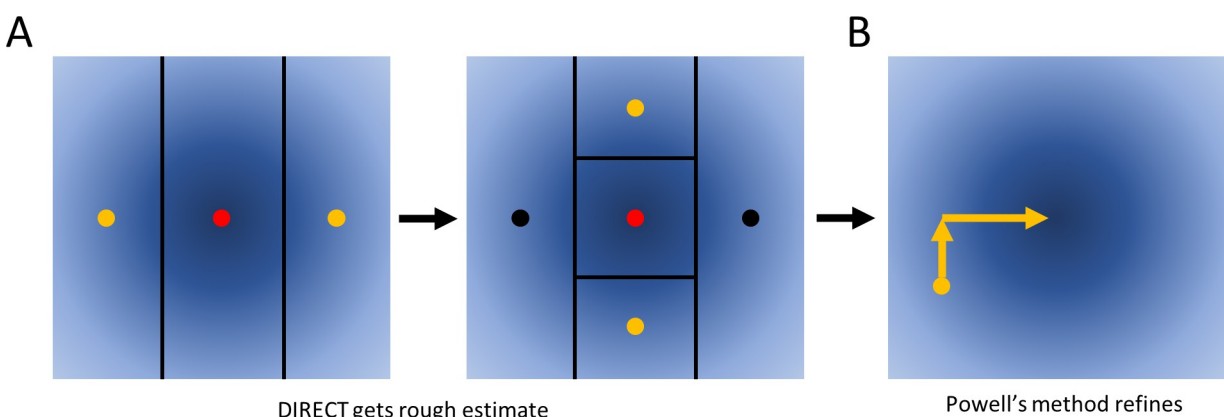

DIRECT gets rough estimate

Powell's method refines

**Fig 7. Illustration of DIRECT and Powell's method.**

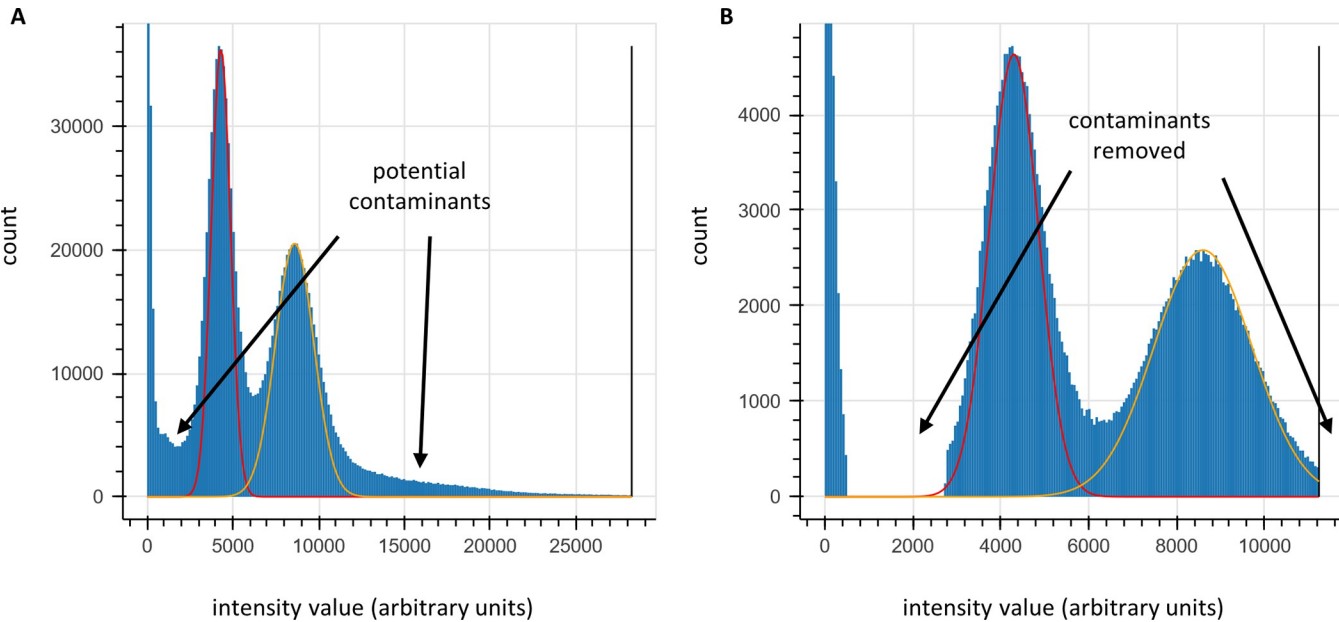

**Fig 8. Determination of fluorescence intensity distribution parameters and filtering of likely contaminants.** (**A**) unaltered histogram of intensity values for a NH2-G{azK}*AG{azK}*| peptide sequencing experiment with superimposed normal distributions (red, fit to peak max ($\mu$) and half-width; yellow, expectation from 2*$\mu$). We typically observe some deviation from a normal distribution that can cause challenges with fitting the distribution, often solved in practice either by fitting the max and peak half-width (as in the red curve) or by trial-and-error using expert judgment. (**B**) clipped data for the same experiment, removing ranges of intensity values likely to be caused by contaminants or signal bleed over from adjacent peaks. Typically, reads are removed from subsequent analyses if any of their intensity values fall in that range.

First, we preprocess our data, either fitting or using expert judgment to determine ranges of fluorescence intensity values that are reasonable for the expected fluorophore counts. An example of this process is shown in Fig 8. This allows us to filter out reads containing improbable intensities that are most likely indicative of a contaminant unrelated to fluorophore signals.

Second, we determine, again through either fitting or expert judgment, the mu and sigma values of every fluorophore color channel in our reads. Doing this through Baum-Welch is unnecessary, as while Baum-Welch is helpful to us in determining parameter values that may interact in unexpected ways, fluorophore intensity distributions should be completely independent of the other parameters so long as we can effectively deconvolute varying fluorophore counts in our analysis. An example of mu, sigma, and background-mu estimation is shown in Fig 8B. We then hold these constant while fitting the other parameters.

Background sigma (the dispersion of the zero-count distribution) is computed automatically through a different mechanism. During signal processing with *sigproc_v2*, the image processing routine in the fluorosequencing data analysis package *plaster SigProc* (https://github.com/erisyon/plaster_v1, as in [22,23]) signal is extracted from the central pixels of a peak, and a local background value is subtracted from the signal. The local background is computed by averaging the values of pixels just outside those used to compute the signal. The median value of these local background values is then used as the background sigma. We have empirically found this to work well in practice.

## Overdetermined systems for low fluorophore counts

For peptides with only one label, many of the parameters are indistinguishable or uninterpretable. In particular, fluorophore destruction and peptide detachment are visibly equivalent when there is only one fluorophore; a fluorophore lost in the middle of a run could just as

easily be a result of either effect. To account for this, when fitting parameters on peptides with one fluorophore, we fix the detachment rate to zero.

Another problematic parameter with peptides of only one label is the missing fluorophore rate, as it is not possible to detect peptides if all fluorophores are missing. When there are two labels on the peptide, we can correct for missing fluorophore biases as described above. But with one, as no peptides with missing fluorophores can be detected, we have no data to ascertain this probability. We therefore hold the missing fluorophore rate to zero in this case as well.

Initial and cyclic blocking rates are difficult to distinguish as well. If the Edman failure rate is allowed to move freely, then the initial and cyclic blocking rates are interchangeable, following an involved formulation:

Consider a peptide with only one fluorophore on the amino acid at position $r$. It can be described by an Edman failure rate of $e$, an initial block rate of $b$, and a cyclic block rate of $c$. Consider an alternative parameterization of the data, where $\kappa_i$ for $i \geq 0$ represents the observed probability of the fluorophore's last cycle being cycle $r+i$. We can relate these two interpretations with the formula:

$$\kappa_i = (1 - b)\binom{i + r - 1}{i}(1 - e)^r e^i (1 - c)^{r+i} \tag{1}$$

The probability at each cycle can be solved for in terms of the preceding cycle:

$$\kappa_{i+1} = \kappa_i e(1 - c)\frac{\binom{i+r}{i+1}}{\binom{i+r-1}{i}} \tag{2}$$

This simplifies to:

$$\kappa_{i+1} = \kappa_i e(1 - c)\frac{i + r}{i + 1} \tag{3}$$

Then the quantity $e(1-c)$ effectively acts together as a single constant, which we can solve for using the ratio between $\kappa_{i+1}$ and $\kappa_i$ for some $i$. A choice of $e(1-c)$ uniquely determines all such ratios, therefore no number of equations for variables $i$ provides any additional information. We can however consider, instead of the ratio, one $i$ on its own. This would allow us to solve for $b$ given a choice of $e$ and $c$. We then have two equations and three unknowns; the system is underdetermined. Because of this, we hold the cyclic blocking rate to zero when only one fluorophore is present on the peptide.

## Bootstrapping

As knowing the precision of a result is important, we provided bootstrapping functionality for both methods by sampling the original data with replacement to create a dataset of the same size. This process can be repeated a user-defined number of times. We also provide, for both methods, the ability to construct a confidence interval using the percentile method of a user-defined size.

## Peptide notation in figures

In subsequent figures and legends, we indicate the amino acid sequences of the peptides being studied as single-letter amino acid codes. For simplicity in this paper, most of the peptides analyzed only have their lysines labeled, and lysines (denoted 'K') are followed by an asterisk, which indicates that they are labeled. Some peptides contained azido-lysine instead of lysine,

denoted '{azK}' which is, like lysine, followed by an asterisk indicating labeling. We notate the status of the N-terminus prior to sequencing, which is either "fmoc-" referring to the fluorenylmethoxycarbonyl protecting group (which is removed prior to sequencing), "ac-" referring to an acetylated N-terminal residue (incorporated to intentionally stop Edman degradation from occurring), or "NH$_2$-" referring to the ordinary unaltered N-terminal state. For simplicity, we also truncate all sequences to the last lysine or azidolysine, as we do not expect amino acids following the last labeled residue to have an impact on the sequencing parameter estimation. We end the sequence notation with a "|" to indicate this omission.

## Experimental fluorosequencing control datasets

We fluorosequenced a number of purified peptides in order to assess parameter estimation on real experimental datasets, including control experiments in which we introduced modifications to the peptides or experimental workflow intended to impact specific error rates. All peptides were synthesized by Genscript and labeled without further purification. The labeled peptides were purified by HPLC and their mass and purity confirmed by LC/MS or SDS page gel purification as described in full in [24]. Briefly, the dye Atto643 was covalently attached to lysine or azido-lysine as appropriate by labeling directly with Atto643 NHS ester or with Atto643 conjugated *via* a polyethylene glycol (PEG)/polyproline tether (termed a Promer) to mitigate dye-dye interactions as in [24]. Peptides were fluorosequenced with a minimum of 40,000 reads using the same sequencing workflow and Total Internal Reflection Fluorescence (TIRF) microscopes denoted Systems A & B in [24].

## Results

### Estimating parameters from simulated datasets as a first test of the algorithms

We evaluated the performance of our modified Baum-Welch algorithm and our DIRECT/Powell's parameter estimation method by analyzing a number of fluorosequencing datasets, some simulated and subsequently others from actual experimental datasets, including ones with controlled manipulation of experimental parameters to test the sensitivity and concordance of the parameter estimators to such effects. We first considered fully simulated datasets, in which the true values of the parameters were known precisely and the (simulated) sequencing process adhered perfectly to the sequencing process implicit in the HMM or Powell's analysis. Although the simulation might not perfectly mimic the real data, evaluating our methods on simulated data provides higher confidence in our techniques; if even on simulated data of a known model our estimators gave bad results we would not expect our methods to be reliable on the more difficult case of real data. Fig 9 plots results for both fitting tools for a two-fluorophore peptide, allowing fitting of all parameters.

Synthetic fluorosequencing reads were generated for peptide NH2-G{azK}*AG{azK}*|, and the simulated dataset was bootstrapped by subsampling with replacement the same number of reads 100 times. Parameters were estimated for each of these bootstrapped subsamples with both methods as described in the text. The box-and-whisker plots represent the distributions of these results, plotting 1st and 3rd quartile +/- max/min observation within 1.5 interquartile range (IQR). Results outside 1.5 times the IQR are considered outliers and are plotted as points. The right facing triangles mark the parameter estimate found if using the non-bootstrapped original data. The dashed black lines indicate the target value that was used in simulation.

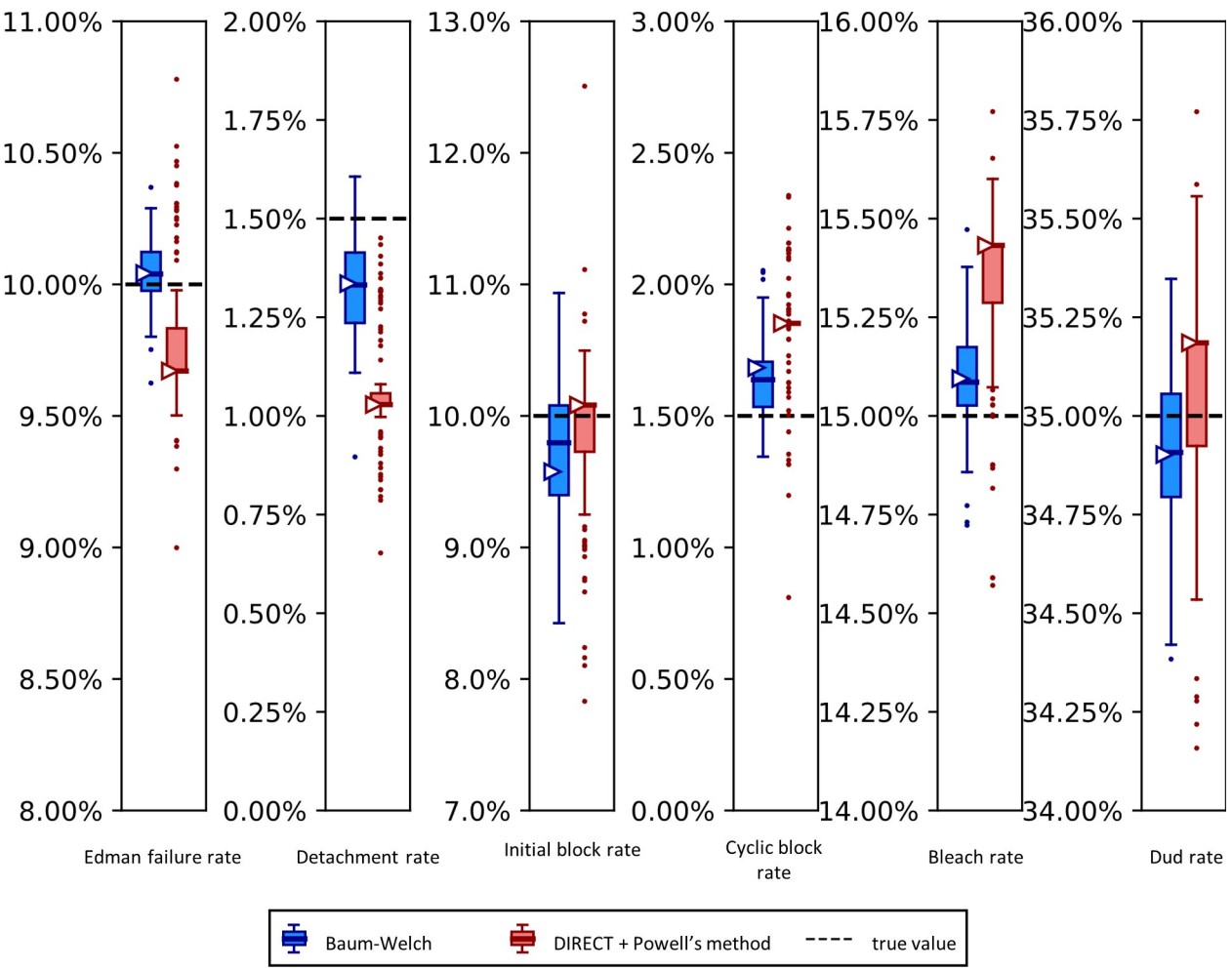

**Fig 9. The Baum-Welch and DIRECT + Powell's methods agree within 0.5% error on simulated data.**

Using both the Baum-Welch and DIRECT followed by Powell's method allowed us to explore potential strengths and weaknesses in both approaches. We saw that our Baum-Welch based approach was able to consistently include the true value within 1.5 times the interquartile range (IQR) of the bootstrapped parameter estimates. For the same synthetic dataset, using DIRECT followed by Powell's method appeared somewhat less reliable—the detachment rate was particularly poorly fit, with the target value lying outside of the range of even the outliers, while the cyclic block rate and the dye loss rate were outside of 1.5 times the IQR but within the more extreme maxima and minima of the outliers. Nevertheless, both methods were moderately close to the target value in most cases, and in all cases, both their primary and median bootstrapped estimates agreed with the true value within 0.5% absolute error.

In order to better understand the effects of variably sized datasets on the size of the confidence intervals given by the Baum-Welch based approach, we next explored this relationship specifically. We anticipated that the scale of the distribution of results would scale with the inverse of the square root of the number of reads in the dataset, as this is the behavior of most distributions in the limit as the number of datapoints rises. However given the complexity of the process being modeled, we thought it wise to verify this by simulation. In Fig 10, we indeed find that the dispersion of the parameter distributions shrinks in the predicted manner, generally converging nicely when analyzing more than 10,000 to 100,000 reads.

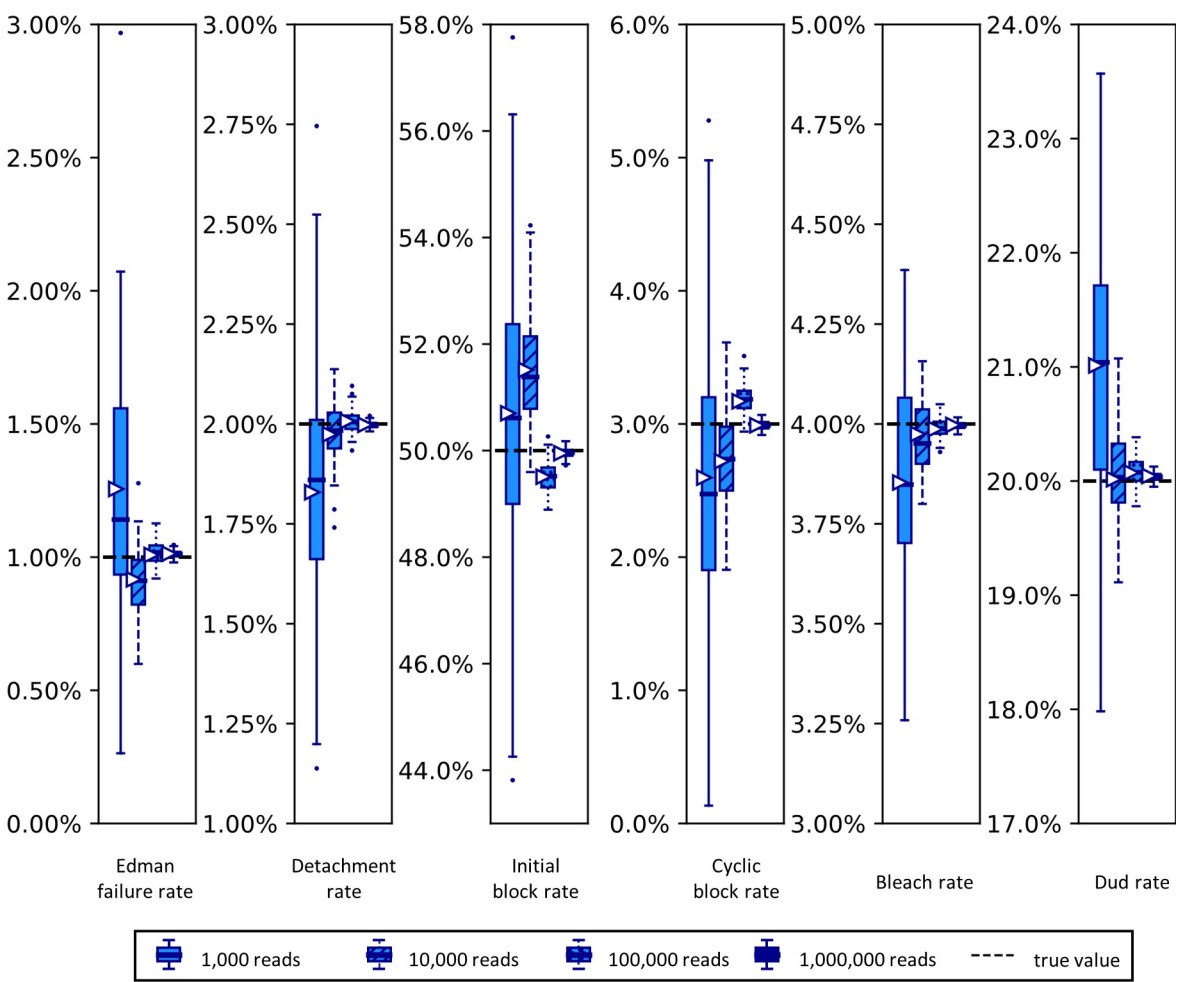

**Fig 10. Simulated datasets with more reads exhibit tighter distributions of parameter estimates.**

Synthetic fluorosequencing reads were created for peptide NH2-G{azK}*AG{azK}*|, simulated the numbers of reads indicated in the graphical legend. These datasets were bootstrapped by subsampling with replacement the same number of datapoints as in the associated simulated dataset, with 100 bootstrapping rounds, and estimating parameters for each of these bootstrapped subsamples with both methods as described in the text. Box-and-whisker plots are defined as in Fig 9.

## Parameter estimation from experimental fluorosequencing datasets

While reproducing the parameters used to generate synthetic data demonstrates the basic validity of the algorithms, they must be able to draw robust estimates from real experimental datasets. We thus collected a series of experimental sequencing datasets designed to intentionally manipulate isolated (where possible) error rates and tested the parameter estimators on these datasets. In all cases, control peptides of known sequence were synthesized, purified, and analyzed, labeling the peptides' lysines (or azido-lysines, as appropriate) with Atto643 dyes on Promer linkers as in [24]. Labeled peptides were then fluorosequenced, collecting a minimum of 30,000 reads. The full details and all experimental procedures and additional control experiments are reported in [24].

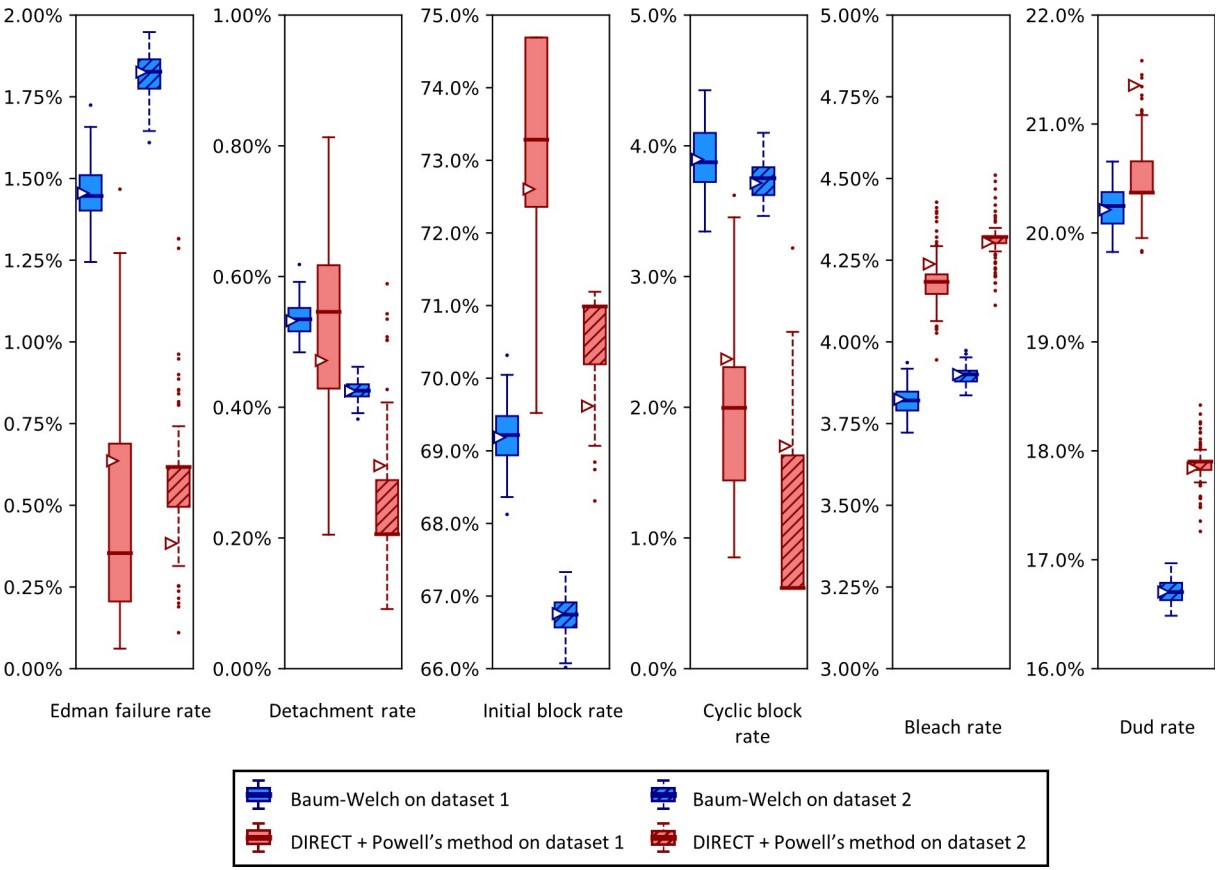

**Fig 11. A comparison of Baum-Welch and DIRECT + Powell's method on experimental sequencing data for a two-fluorophore peptide shows general agreement between the methods.**

Importantly, sequencing a known peptide labeled with two fluorophores allows us to fit all six free parameters simultaneously. We demonstrate this for both fitting methods with two independently collected experimental datasets in Fig 11. In general, the methods estimated parameters within a few percent of each other. However, we noted a tendency for DIRECT followed by Powell's method to give results on the original dataset (triangle markers) distant from the median results of the bootstrapped data (box plot midlines), in some cases even outside the first-to-third quartile range. Its distributions also tended to be much broader than those given by Baum-Welch, suggesting less confidence in its estimates. However an alternative interpretation could be considered: the wider distribution of DIRECT followed by Powell's method allowed for overlapping distributions across the replicate datasets in every case except for the missing fluorophore (dud) rate, while Baum-Welch produced distributions that fail to overlap completely for not only the missing fluorophore rate, but also the initial block rate, and the Edman failure rate had only the slightest overlap. Thus, DIRECT+Powell's appeared more consistent with the scale of parameter variation observed across experimental replicates.

Fluorosequencing datasets for peptide NH2-G{azK}*AG{azK}*| were collected one day apart by the same researcher on the 21st (dataset 1 with 40,181 reads) and the 22nd (dataset 2 with 71,823 reads) of November 2022. The original datasets were bootstrapped by subsampling with replacement the same number of datapoints as in the original dataset 100 times, and fitting on each of these bootstrapped subsamples with both methods as described in the text. Box-and-whisker plots are defined as in Fig 9.

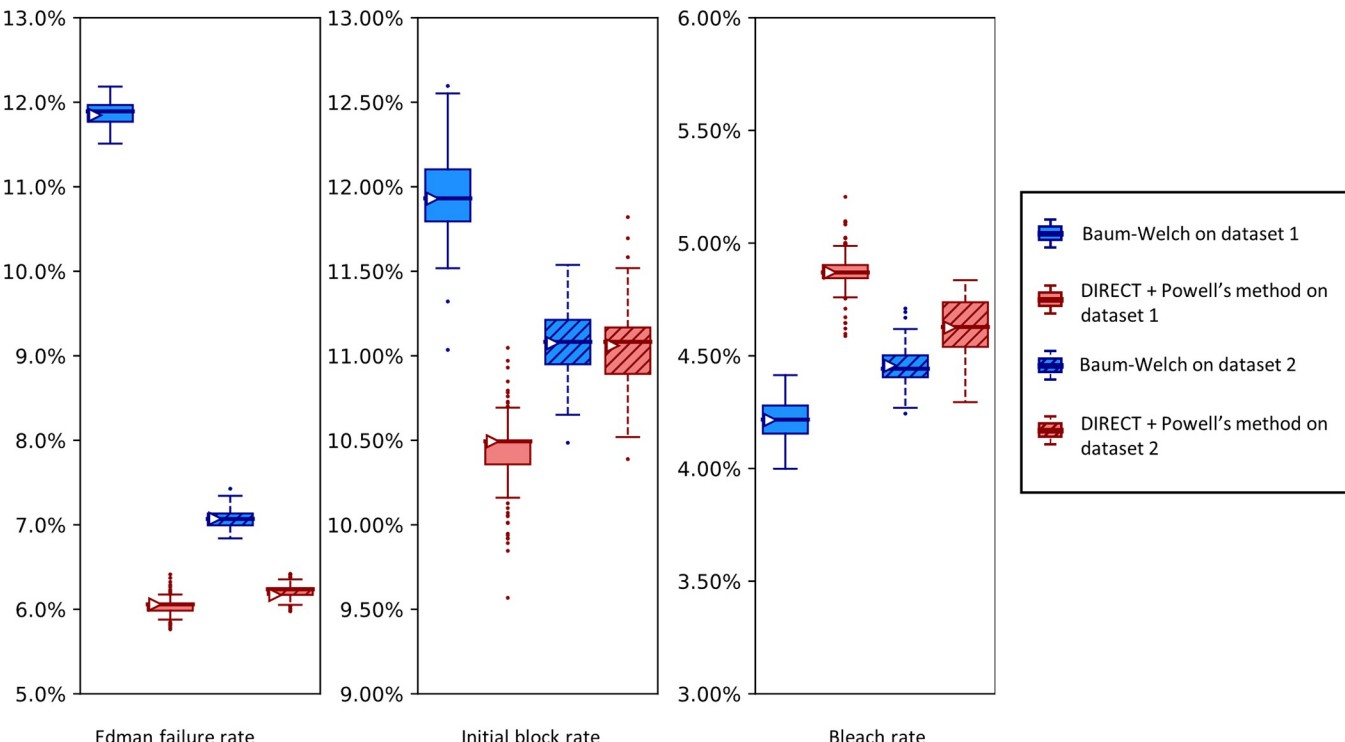

**Fig 12. A higher Edman failure rate is observed for a proline-containing peptide, with general agreement between the estimation methods.**

We also observed an unusual tendency in Figs 9 and 11 for DIRECT with Powell's method to produce distributions of fits from bootstrapped data with large numbers of outliers, with medians sitting on top of the first or third quartile, and in one case (cyclic block rate in Fig 11) the median even appears to be identical to the minimum. We do not have an explanation for this phenomenon.

In the course of other studies, we noticed varying Edman cleavage efficiencies at proline residues, which had been purported to sequence less efficiently in historic Edman degradation experiments [25–28]. Initial fluorosequencing studies saw little effect stemming from proline [13], but subsequent studies showed less efficient cleavage, an effect we were able to trace to the length of incubation time in trifluoroacetic acid (TFA) [24], a key reagent for Edman degradation whose usage had been reduced while optimizing the sequencing workflow. This effect thus provides a means to experimentally manipulate the Edman failure rate and test the response of the parameter estimators.

Fig 12 presents such an analysis, plotting the results of analyzing two independently collected fluorosequencing datasets for the same proline-containing peptide with a single fluorophore in the third position. Results for Baum-Welch differed considerably, while the DIRECT with Powell's method remained more consistent. In particular, the Baum-Welch method's discrepancy in the Edman failure rate between the two experiments, predicting values of approximately either 12% or 7%, is concerning, as while it is possible that the true value did vary by this amount, the DIRECT/Powell's method shows higher concordance between the experiments.

Here, we analyze two experimental fluorosequencing datasets for peptide fmoc-APK*| collected by the same researcher on the 11th (dataset 1 with 27,783 reads) and 28th (dataset 2 with 34,380 reads) of November 2022. The original datasets were bootstrapped by subsampling

with replacement the same number of datapoints as in the original dataset 100 times, and fitting on each of these bootstrapped subsamples with both methods as described in the text. Box-and-whisker plots are defined as in Fig 9.

As a comparison with the proline containing peptide in Fig 12, in Fig 13 we present results for a peptide that has no prolines. As expected, the no-proline case has a much higher Edman cleavage efficiency (1—Edman failure rate) than for the case of the proline-containing peptide. We also see a significantly higher initial block rate and dye loss rate, the former of which may relate to less efficient cleavage at the glycine residue.

The peptide fmoc-GAK*| was sequenced on December 1st, 2022, and the original dataset of 67,936 reads was bootstrapped by subsampling with replacement the same number of datapoints as in the original dataset 100 times, and fitting on each of these bootstrapped subsamples with both methods as described in the text. Box-and-whisker plots are defined as in Fig 9.

We would expect an acetylated, or blocked, peptide to be fit with either a very high Edman failure rate or initial block rate, or perhaps both. Fig 14 confirmed this prediction, with initial block rates strictly above 91% for all bootstrapped results with either dataset and either fitting technique. While these results raise our confidence in both techniques, they are lower than anticipated. We suspect that contaminants captured by accident in this dataset are responsible for this behavior. We also conjecture that the 40% range of discrepancies between datasets in the Edman failure rates arose because of low signal and increased sensitivity to contaminants due to the high initial block rate.

Here, we examine two independent fluorosequencing datasets for ac-A{azK}*|, an N-terminally acetylated peptide, i.e. one for which the N-terminus is covalently blocked and not

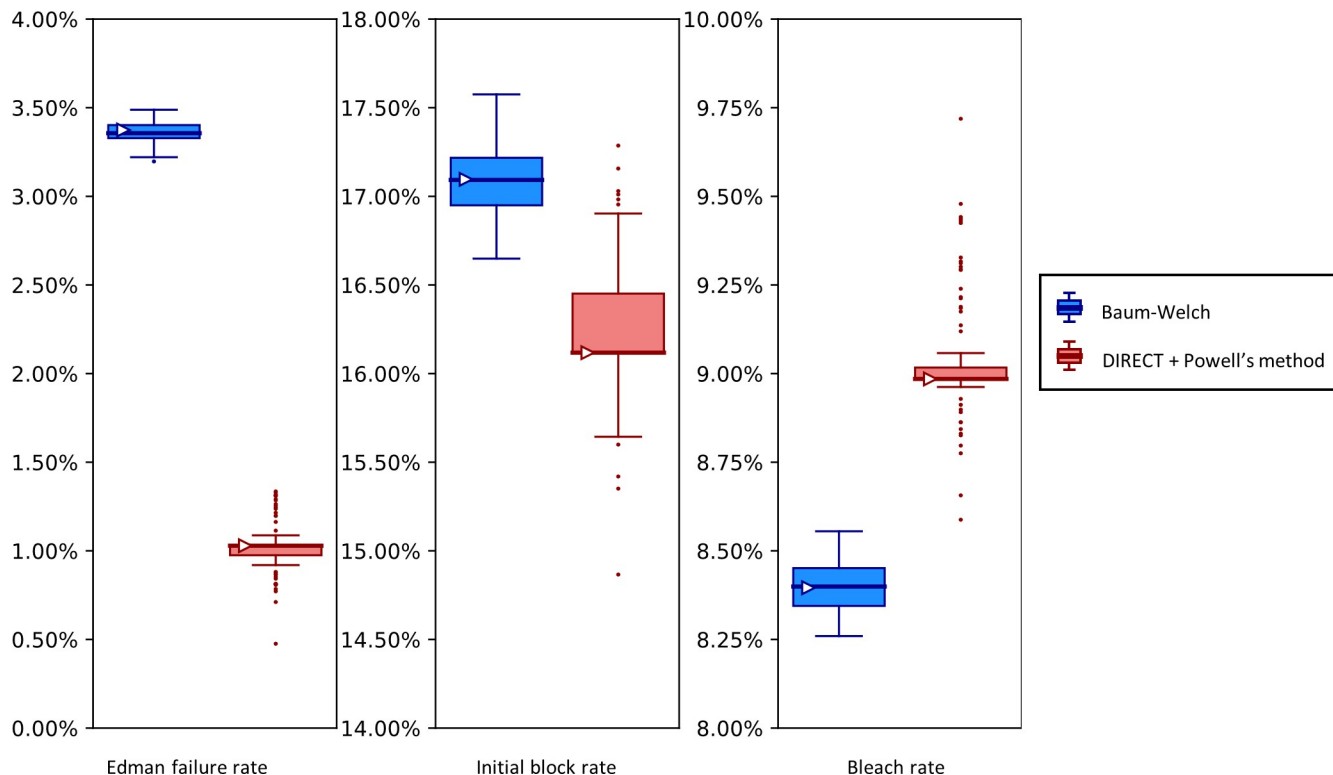

**Fig 13. Analysis of experimental fluorosequencing data from a peptide similar to that in Fig 12 but containing no proline residues exhibits lower Edman failure rates and shows agreement between the methods.**

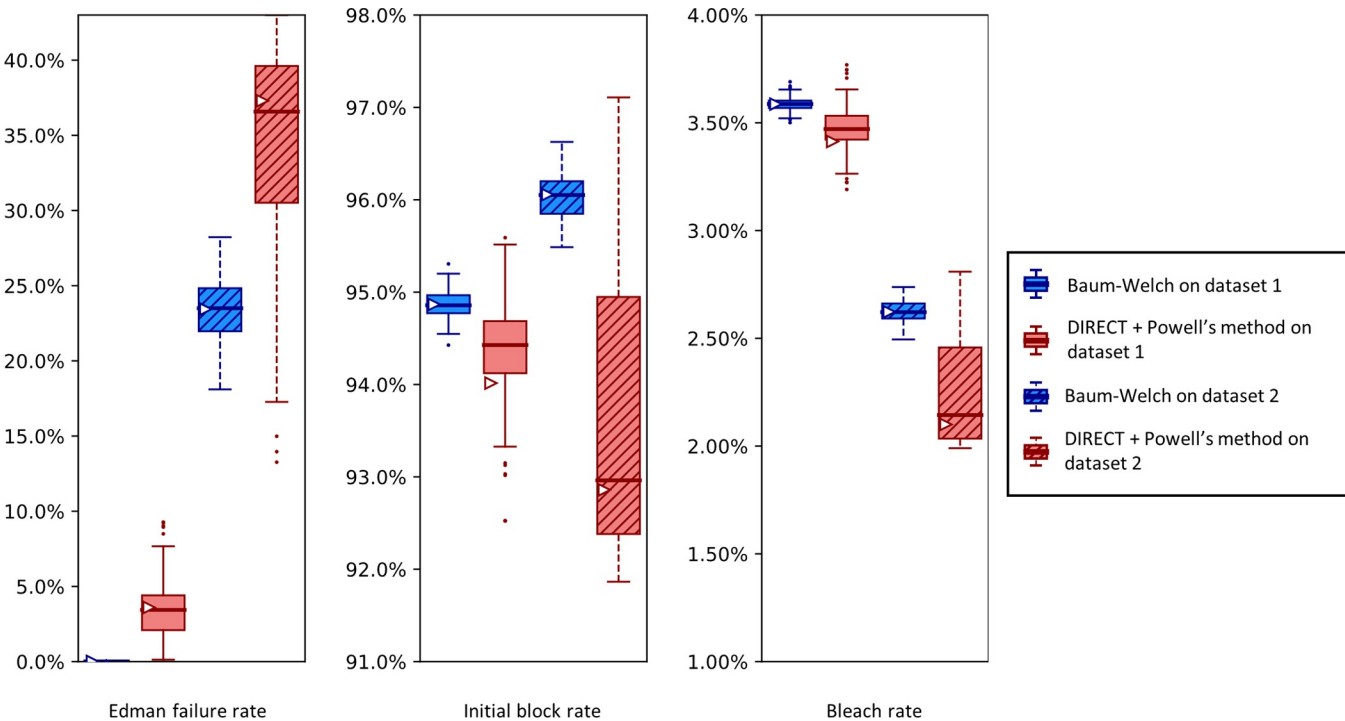

**Fig 14. Both parameter estimators correctly recognize high initial block rates (>91%) in an intentionally N-terminally blocked peptide.**

sequenceable by Edman chemistry. The datasets were collected on the 15th (dataset 1 with 43,606 reads) and 17th (dataset 2 with 42,417 reads) of November 2022 by the same researcher. The original datasets were bootstrapped by subsampling with replacement the same number of reads as in the original dataset 100 times, and fitting on each of these bootstrapped subsamples with both methods as described in the text. Box-and-whisker plots are defined as in Fig 9.

In order to test whether other parameters were being correctly estimated, we also manipulated the incubation time in trifluoroacetic acid (TFA), which should directly relate to Edman failure rates, for each of 6 independent peptides. TFA is a key reagent for Edman degradation (Fig 1), but also one known to inactivate dyes with sufficiently long incubations [13]. In our initial fluorosequencing paper, we performed 15 or 30 minute incubations [13], but subsequent tests revealed that incubation times could be shortened with little loss of cleavage efficiency [24]. Thus, we wished to test the minimum TFA incubation times and explore the intrinsic tradeoff between cleavage efficiency and dye loss rates.

Fig 15 plots how the various estimated error rates changed on experimental fluorosequencing data collected with 4 different TFA incubation times during the Edman degradation steps. As expected, reducing TFA incubation time to near zero increases the Edman failure rate for all peptides studied, but with as little as 4–6 minutes, all peptides with the exception of the proline containing peptide exhibit low Edman failure rates, hence reasonably efficient cleavage efficiency (Fig 15A). The effect of longer TFA incubation time on the initial N-terminal blocking rate is less clear, with large effects primarily noted on the proline containing peptide, suggesting there may be some tradeoff by the parameter estimators in assigning weight to the Edman cycling versus initial block in that specific case (Fig 15B). Also as expected, longer exposure to TFA results in higher dye loss rates (Fig 15C), consistent with our prior observations.

We tested minimum times required for TFA incubation by sequencing six different peptides with four different lengths of time of TFA exposure. Every dataset used in these plots has

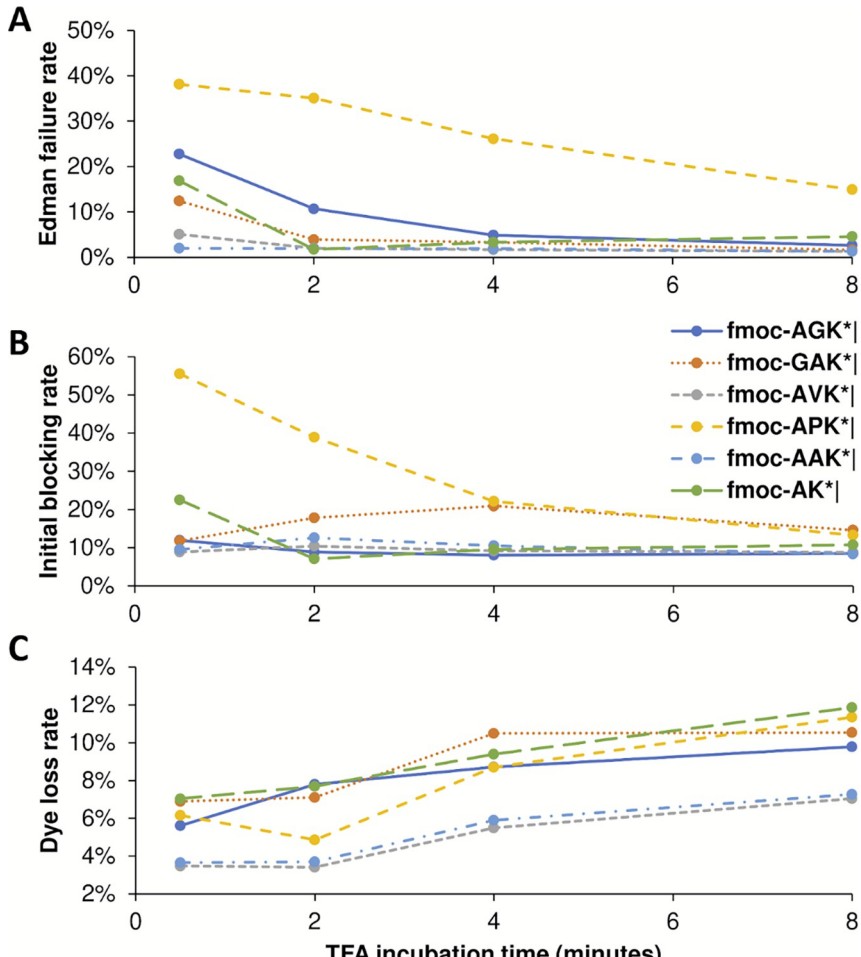

**Fig 15. Longer TFA incubation times reduce the Edman failure rate but increase the dye loss rate.**

at least 34,000 reads, and all but three have more than 50,000 (see Zenodo repository for precise counts). (**A**) The Edman failure rate decreases with longer TFA incubation time, which matches theoretical expectations. (**B**) The initial block rate does not follow a clear pattern. (**C**) The dye loss rate goes up with longer TFA incubation time, which matches theoretical expectations.

## Discussion

A reason to use two parameter estimation methods was to measure their agreement on real data; in Fig 11 we unfortunately see wide discrepancies between the two methods. This was not observed for the simulated data in Fig 9. We observed less severe modeling discrepancies for the real data plotted in Figs 12–14 as well. We believe these discrepancies are a result of the contaminants and other unknown factors that are present in data. This could cause slightly incorrect results for both models, but they may react differently in the case of an incomplete model.

Some of the more unusual results may be caused by errors in sample preparation or data collection. Across all real datasets (Figs 11–15), we observe variations in the initial block rate that do not follow a clear or consistent pattern. We suspect that some step in sample

preparation or data collection is ultimately responsible, but we have as yet been unable to determine the root cause.

For other parameters, such as the dye loss rate, there are small but statistically significant fluctuations between experiments, not caused by modeling differences between the methods. As with the larger discrepancies in the initial block rate, inconsistent sample preparation or data-collection practices could be responsible. However other interpretations are possible. Indeed, some amount of inconsistency should be expected, as any number of factors can change, including variations in reagents or imaging quality. It is also possible that our data analysis practices are at fault. We have observed random fluctuations such that all intensity values may be a small amount brighter or dimmer at each cycle of our experiment. Applying the same distributional cut-offs and distributions of intensity across all time steps may then introduce biases in the data that is kept and in how it is analyzed, resulting in small fluctuations for multiple parameters. This is a possible avenue of future exploration.

We can also ask which of our methods is better. With a clear winner between Baum-Welch and Powell's method, we could reduce future explorations to the better method. Baum-Welch provides more accurate results for simulated data; in Fig 9 both the estimates on the original data and the medians of the bootstrapping results are better for five out of the six parameters with the Baum-Welch algorithm than with the Powell's method-based results. Baum-Welch also tends to have tighter confidence intervals and more consistent distributional shapes than what we see with Powell's method. Baum-Welch mostly appears to be the better technique, but questions remain, in particular the discrepancy between the two replicates shown in Fig 12 for the Edman failure rate for Baum-Welch but not for Powell's method.

Finally, our new parameter estimation techniques prompted us to compare our experimental results to previous, though less precise, results of fluorosequencing in earlier publications [13]. A number of advancements have been made which have improved these error rates [24]. In 2018 the Edman failure rate was measured at around 6%. Edman failure rates now appear to be around 1% or 2% for most residues and for much shorter TFA incubation times, following extensive optimization of the reaction conditions [24]. A consequence of the shorter TFA incubations, however, is that error rates at prolines remain higher, as in Fig 12. Dye loss rates and peptide detachment rates were estimated at 5% in 2018; while current results suggest an unchanged peptide detachment rate, the dye loss rate has been brought below 1%, perhaps to 0.5%. The missing fluorophore rate appears to have risen from 7% to 20% in this time-frame, and experiments may be needed to investigate the source of this rate.

## Conclusion

We have developed and tested two computational approaches for estimating key parameters governing protein fluorosequencing experiments, which should provide guidance for refining experimental protocols as well as improve the quality of peptide classification performed on the resulting datasets. One approach introduces modifications to the Baum-Welch algorithm that allow it to better isolate sources of error from each other and to then compute weighted MLEs at each step of the Baum-Welch Expectation Maximization process. The other approach minimizes the RMSE of the dye-track counts between simulations from the parameters and the provided dataset. Both methods seem promising and are generally concordant, and thus provide important checks on each other. We anticipate these approaches will be generally useful for ongoing efforts at improving the fluorosequencing experimental pipeline, including better identifying and mitigating mis-sequenced peptides and fluorescent contaminants, as well as for developing improved computational methods for interpreting fluorosequencing datasets.

## Supporting information

**S1 Appendix. Technical details of Baum-Welch.**
(PDF)

**S2 Appendix. Technical details of error type isolation.**
(PDF)

**S3 Appendix. Technical details of maximum likelihood estimation.**
(PDF)

**S4 Appendix. Technical details of bias correction.**
(PDF)

## Acknowledgments

The authors gratefully acknowledge Jagannath Swaminathan, Eric Anslyn, and Daniel Weaver for helpful guidance and discussion throughout the course of this project.

## Author Contributions

**Conceptualization:** Matthew Beauregard Smith, Kent VanderVelden.

**Data curation:** Thomas Blom, Heather D. Stout, James H. Mapes, Tucker M. Folsom, Christopher Martin, Angela M. Bardo.

**Investigation:** Matthew Beauregard Smith, Kent VanderVelden.

**Software:** Matthew Beauregard Smith, Kent VanderVelden, Thomas Blom.

**Supervision:** Angela M. Bardo, Edward M. Marcotte.

**Visualization:** Matthew Beauregard Smith.

**Writing – original draft:** Matthew Beauregard Smith, Kent VanderVelden, Heather D. Stout, Edward M. Marcotte.

**Writing – review & editing:** Matthew Beauregard Smith, Edward M. Marcotte.

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
