## [Decision Letter · Decision Letter 0]

2 Feb 2024

Dear Smith,

Thank you very much for submitting your manuscript "Estimating error rates for single molecule protein sequencing experiments" for consideration at PLOS Computational Biology.

As with all papers reviewed by the journal, your manuscript was reviewed by members of the editorial board and by several independent reviewers. In light of the reviews (below this email), we would like to invite the resubmission of a significantly-revised version that takes into account the reviewers' comments.

We cannot make any decision about publication until we have seen the revised manuscript and your response to the reviewers' comments. Your revised manuscript is also likely to be sent to reviewers for further evaluation.

Sincerely,

Alexandre V. Morozov, Ph.D.

Academic Editor

PLOS Computational Biology

William Noble

Section Editor

PLOS Computational Biology

Reviewer's Responses to Questions

**Comments to the Authors:**

Reviewer #1: The authors present a methodology to analyze SMPS read data and estimate the error rate of reads. To estimate the error rate, from different sources, they adapted the Baum-Welch algorithm. The method was evaluated on both simulated and experimental data.

While I think the method is going to be very valuable for the field, the current manuscript mainly lacks in showing its application potential. Could the authors address the points outlined below?

1. As the authors indicate, there are discrepancies between the simulated and experimental data and estimates of the error rates. In the worst case, this suggests that the simulated data is incorrect and cannot be used to evaluate methods presented here. Is there any procedure the authors went through to validate the simulated data? In the discussion, currently, the question of the cause of this discrepancy is kept open. I think the authors should further investigate this or leave the simulation part out of the manuscript. As it is not clear to me whether this data is actually valid and can be used.

2. To me it seems that the authors mainly focused on the stability/reproducibility of their error estimates. To me it is unclear if the authors validate these numbers to be correct in their manuscript. Also, I do not see a clear application for the current state of the model, could the authors go more in-depth on how these error-estimates should be used? It would also be great if the authors show an application of the model.

3. The authors mention that their implementation of Baum-Welch is more robust to overfitting, but to me this is not apparent from the manuscript.

Reviewer #2: The authors provided an advanced method to estimate the parameters or error rates for single-molecule protein sequencing. This method extended the Bamum-Welch algorithm to the previous algorithm Whatprot model which is based on the HMM model to perform peptide classification on florosequencing data. The Bamum-Welch algorithm is adapted to make use of the forward-backward algorithm to maximize the likelihood by finding the unknown parameters in the HMM model. The authors demonstrated the high accuracy of parameter estimation on simulated by adopting the Bamum-Welch algorithm in the HMM model. Meanwhile, the authors provided a second option using DIRECT and Powell’s method to reduce the RMSE which also has been proven on simulation and real datasets.

The paper showed a clear idea about the method and solid results to support the application. The authors gave comprehensive mathematical model explanation and detailed proof. Overall, it’s well-prepared to be accepted. Here are only some minor suggestions.

1) The main figure 1 depicts the essential steps of single-molecule protein sequencing and labels the potential error rates of the steps, which are the parameters that need to be estimated by the HMM model. The caption gives a detailed technical explanation of Figure 1 from the chemistry and sequencing aspects. However, there are lack of demonstration to build the mathematical model with real steps in fluorosequencing. For example, where/which steps are the hidden states generated from? What does transition_probability/ emission_probability represent in those steps? What are the observations? The mathematical notations representations are essential to help readers to understand how to build the model.

2) The same problem existed in Figure 2, especially, since there is no clear explanation/notations to demonstrate how the hidden Markov model applies for fluorosequencing. Could you please give a simple example to clearly show how the HMM model matches the SMPS?

3) In the paper, there are multiple times mentioned “as in Chapter 2”, for example, line 158, line 171, line 1032… Is there any missed literature that needs to be cited?

4) In line 170, “illustration from Figure 2.4 to include N-terminal blocking ”, I could not find Figure 2.4, please correct the Figure citation.

**Have the authors made all data and (if applicable) computational code underlying the findings in their manuscript fully available?**

Reviewer #1: Yes

Reviewer #2: Yes

PLOS authors have the option to publish the peer review history of their article (what does this mean?). If published, this will include your full peer review and any attached files.

Reviewer #1: No

Reviewer #2: No
---

## [Decision Letter · Decision Letter 1]

17 Jun 2024

Dear Smith,

We are pleased to inform you that your manuscript 'Estimating error rates for single molecule protein sequencing experiments' has been provisionally accepted for publication in PLOS Computational Biology.

Best regards,

Alexandre V. Morozov, Ph.D.

Academic Editor

PLOS Computational Biology

William Noble

Section Editor

PLOS Computational Biology

Reviewer's Responses to Questions

**Comments to the Authors:**

Reviewer #1: The authors have substantially improved their manuscript and have addressed my concerns.

Reviewer #2: No more questions about the revised manuscript.

**Have the authors made all data and (if applicable) computational code underlying the findings in their manuscript fully available?**

Reviewer #1: Yes

Reviewer #2: Yes

PLOS authors have the option to publish the peer review history of their article (what does this mean?). If published, this will include your full peer review and any attached files.

Reviewer #1: No

Reviewer #2: No

---

## [Editor Report · Acceptance letter]

29 Jun 2024

PCOMPBIOL-D-23-01562R1 

Estimating error rates for single molecule protein sequencing experiments

Dear Dr Smith,

I am pleased to inform you that your manuscript has been formally accepted for publication in PLOS Computational Biology. Your manuscript is now with our production department and you will be notified of the publication date in due course.

With kind regards,

Lilla Horvath
